# Effect Mechanism and Simulation of Voids on Hygrothermal Performances of Composites

**DOI:** 10.3390/polym14050901

**Published:** 2022-02-24

**Authors:** Zhu Liu, Yongpeng Lei, Xiangyang Zhang, Zhenhang Kang, Jifeng Zhang

**Affiliations:** Key Laboratory of Advanced Ship Materials and Mechanics, College of Aerospace and Civil Engineering, Harbin Engineering University, Harbin 150001, China; liuzhu0618@hrbeu.edu.cn (Z.L.); zxy121382021@163.com (X.Z.); kangzhenhang@hrbeu.edu.cn (Z.K.)

**Keywords:** void content, hygrothermal aging, micromechanics, failure mechanism

## Abstract

Voids are comment defects generated during the manufacturing process and highly sensitive to moisture in the hygrothermal environment, which has deleterious effects on the mechanical performances. However, the combined impact of void content and water-absorbed content on mechanical properties is not clear. Based on the random sequential adsorption algorithm, a microscale unit cell with random distribution of fibers, interfaces and voids was established. The quantitative effects of voids content on strength and modulus under the loading of transverse tension, compression and shear were investigated by introducing a degradation factor dependent on water content into the constitutive model, and the different failure mechanisms before and after hygrothermal aging were revealed. Conclusively, before hygrothermal aging, voids induce the decrease in mechanical properties due to stress concentration, and every 1% increase in the void content results in a 6.4% decrease in transverse tensile strength. However, matrix degradation due to the absorbed water content after hygrothermal aging is the dominant factor, and the corresponding rate is 3.86%.

## 1. Introduction

Voids are the most common type of defects induced by the residual air during the manufacturing process of composites [1], which significantly affect the hygrothermal aging performances of composites by altering the stress field and moisture field [2,3]. Particularly, void defects are highly sensitive to the moisture under the hygrothermal environment, and they can further decrease the matrix-dominated properties, which can ultimately reduce the service life of composite structures [4,5]. Thus, it is crucial to understand the effect of voids on hygrothermal aging performances and reveal their failure mechanism.

Various characterization techniques were reported for identifying the microscopic void structures and voids content, including densities measurement method [6], optical image analysis [7], infrared thermography [8] and micro X-ray computed tomography [9]. However, these techniques are either unreliable due to inherent testing errors or complex operations at high costs. Thus, many researchers employed the finite element method (FEM) to study the behavior of composites with voids [10,11,12]. Based on the FEM model and analytical model, a parametric study was conducted by Huang et al. [13] to reveal the effects of voids geometry and distribution on the elastic constants. Nikopour et al. [14] developed a representative volume element (RVE) with the order voids’ arrangement to investigate the effect of voids on an estimation of effective transverse properties of composites. Vajari et al. [15] further developed the RVE model with random distribution of fibers and voids, and revealed that the presence of voids significantly reduces the strength of composites. A regression model was presented by Chen et al. [16] to predict the effects of voids on the strength and modulus, based on FEM results. RVE models with different distribution patterns and shapes for voids were established by Wang et al. [17], and the effects of voids on the transverse tensile strength considering thermal residual stress were analyzed. A combined micro-scale and meso-scale methodology were proposed by Mehdikhani et al. [18] to examine the effect of intra-laminar voids on the evolution of cracks. Hyde et al. [19] vastly discussed the RVE model with a single void and a stress concentration factor was introduced to quantify the effect of void volume fraction, void orientation, shape density and associated void defects to illustrate mechanical properties of the composites. Though voids are highly sensitive to moisture in the hygrothermal environment, only several literatures reported the effects on moisture absorption performances. Gueribiz et al. [20] proposed analytical solutions by solving unit cell problems on RVE to determine the effective diffusivities of composites, including the Porous Matrix model, Four-phase model and Self-consistent model, and investigated the effects of voids on the moisture diffusivities of composites. Bourennane et al. [21] developed an RVE model with an elliptical-shaped void, considering two types of closed and open voids, and investigated the effects of geometric configuration and volume fraction on the moisture diffusion process in the damage polymer matrix.

PA6 matrix can absorb about 10 wt% water [22], and the absorbed water has deleterious effects on the mechanical performances of PA6-based composites [23,24,25]. It is necessary to understand the combined impact of hygrothermal aging and voids defects on the water absorption properties and mechanical performances for PA6-based composites. Although extensive studies involved the effects of voids on mechanical properties, these studies focused on composite laminates without hygrothermal aging. In fact, the failure mechanism of composite induced by the presence of voids before and after hygrothermal aging is totally different. Thus, to reveal the failure mechanism, the degradation of mechanical properties dependent on the absorbed water content should be introduced into the constitutive model.

## 2. Computational Model Considering Hygrothermal Aging

### 2.1. RVE Model with Voids

The presence of void defects has deleterious effects on the service performance of composites. Several manufacturing parameters including molding pressure, curing temperature and resin viscosity can induce the formation of void defects in the matrix, as shown in Figure 1a. In composites, voids are found in different volume fractions, shapes and sizes. In this study, the main purpose is to reveal the effect of voids content on the water absorption behavior and mechanical performances. Thus, the void generation algorithm fulfil the following terms: (1) voids only exist in the matrix, and voids are approximately circular in shape; (2) the void distribution is random, the positions of voids are automatically generated using the random sequential adsorption (RSA)-based algorithm programmed in MATLAB (2018a) [26,27]; (3) as shown in Figure 1b, void size distribution is in accordance with the Weibull distribution, and the values of scale and shape are 15 and 1.5, respectively; (4) there is no intersection between voids and fibers, and void contents of 0%, 1%, 2%, 3%, 4% and 5% are considered in this work, as shown in Figure 1c.

### 2.2. Constitutive Models in Water Diffusion Process

Moisture absorption testing is conducted to measure the masses of PA6 resin and CF/PA6 composites, according to the standard of ASTM D5229. Water absorption curves are plotted by recording the masses of samples over time. Based on Fick law, the maximum moisture content (*M_i_*) and the diffusivity coefficient ($D_{i}$) of component materials can be obtained except for voids. Based on the thermodynamic laws for air, the water content absorbed by voids can be determined by the following equation [21]: $M_{v}=0.622P_{V}/\left(P_{atm}-P_{V}\right)$, and $P_{V}$, $P_{atm}$ are the saturation vapor pressure and the atmospheric pressure, respectively. In this work, it is assumed that the diffusivity of voids is one order of magnitude higher than that of the matrix, due to its faster water absorption rate. More details of water absorption parameters are listed in Table 1.

The mechanical properties of the matrix and the interface decrease significantly with the water content increasing under the hygrothermal environment. Thus, an improved traction-separation cohesive law is developed to study the debonding failure under different load conditions, by introducing degradation factors for strength, modulus, and fracture energy. Carbon fibers are modeled as transversally isotropic and linear elastic. The yielding of the PA6 matrix is described by the extended linear Drucker-Prager criterion [28], which considers different yielding strengths in tensile, compression and shear behaviors:(1)$$\left\{ \begin{array}{l} F=t-p\tan\beta -d=0\\ t=\frac{1}{2}q\left[1+\frac{1}{k}-\left(1-\frac{1}{k}\right){\left(\frac{r}{q}\right)}^{3}\right] \end{array}\right.$$
where *p* is the hydrostatic stress, *q* is the Mises equivalent stress, *r* is the third invariant of deviatoric stress, β is the slope of the linear yield surface in the *p–t* stress plane, *d* is the cohesion of the material and *k* is the ratio of the yield stress in triaxial tension to the yield stress in triaxial compression.

Moreover, the ductile criterion is employed to predict damage onset of the PA6 matrix by assuming the equivalent plastic strain as a function of stress triaxially η(η=−p/q). Here, η takes the value of 1/3, −1/3, 0 under the uniaxial tension, uniaxial compression and pure shear, respectively. Besides the yield criterion, the effect of water content on the mechanical properties is also considered in this work. Experimental hygrothermal aging has quantified the degradation rate of PA6 mechanical properties with respect to the absorbed water content [23], and the retention percentage of strength and modulus can be described as follows:(2)$$\left\{ \begin{array}{l} \sigma_{t}=37+\frac{100-37}{1+\exp\left[1.08\;\times \;\left(M_{t}-2.76\right)\right]}\\ E_{t}=15.51+\frac{100-15.51}{1+\exp\left[1.66\;\times \;\left(M_{t}-2.81\right)\right]} \end{array}\right.$$ where $\sigma_{t}$ and $E_{t}$ are the retention percentage of tensile strength and elastic modulus after hygrothermal aging, *M_t_* is the water-absorbed content.

The interface behavior is described with the cohesive element model defined in terms of bi-linear traction-separation law. Similarly, the relationship between the mechanical properties and the water content is assumed to be the same as that of PA6 matrix. The cohesive elements between the neighboring matrix and fiber elements are automatically inserted in the Abaqus. All mechanical parameters used in this work are listed in Table 2.

## 3. Experimental Verification

To validate the accuracy of the FEM model, macro-mechanical testing is conducted by comparing the strength, modulus and failure model. Firstly, CF/PA6 prepregs are prepared by a melt impregnation technique, and CF/PA6 composite laminates are fabricated using a high temperature compressing molding technique. The average fiber volume fraction is 30 vol% determined by the ablation method according to ASTM D2584, and the voids content is about 0.13% determined by optical image analysis. Then, the transvers tensile, compression and shear test are also conducted according to the ASTM D3039, D6641 and D5379, respectively. Five replicated samples are used for mechanical testing, and the corresponding strength and modulus are the average of the five tests. The strength and modulus with standard deviation in the transvers tensile, compression and shear test are listed in Table 3.

The stress–strain curves of the RVE model without voids under the transverse tension, transverse compression and transverse shear are shown in Table 2. The blue images in the first row are the equivalent plastic strain (PEEQ) nephogram of the whole RVE model, which indicates the plastic deformation under the three loading modes. The white images in the second row are the stiffness degradation (SDEG) nephogram of cohesive elements, which indicates the damage revolution of the interface during the different load process. The three columns of the images are corresponding to the onset of interface failure (“1”), the peak load (”2”) and the end of loading (”3”), respectively. The mechanical performances, including strength and modulus determined by FEM, are also compared with the experimental data to verify the accuracy of numerical simulation.

Under transverse tension, the fiber/matrix is the main reason for the failure process. Figure 2a shows that the initial damage occurs at the fiber poles along the loading direction and gradually evolved to the fiber/matrix debonding. Remarkably, the stress concentration arises in the regions where the clustering degree of fibers is relatively higher, and these regions are susceptible to the interfacial debonding. Then, the matrix at the vicinity of interfacial debonding undergoes plastic deformation and accumulation damaged until ultimate failure. Final interfacial debonding at different locations are linked by matrix cracks, and the final damage cracks of RVE are formed perpendicular to the loading direction. Under transverse compression, the final failure is primarily through the shear bonds of the matrix. As shown in Figure 2b, the initial failure is induced by the interfacial debonding and then a plastic shear band is formed oriented at about 56° with respect to compression direction. The shear bonds further develop along this direction, accompanied by interfacial debonding to jointly final failure of composites. Under transverse shear, the failure mechanism is dominated by interfacial debonding and matrix yielding. The initial fracture is triggered by interfacial debonding, similarly to the transverse compression case. As the matrix holds progressively shear loads, an obvious plastic band is formed and cracks grow until the final failure, as shown in Figure 2c.

The strength and modulus determined from this work agree well with the experimental results, and the relative errors are less than 10% except for the transverse shear modulus. The ultimate damage mode is quite similar to the experimentally observed results [29,30,31].

Thus, the present computational framework is effective in predicting mechanical properties and failure modes of composites, as well as composites with void defects.

## 4. Results and Discussion

### 4.1. Water Absorption Behavior

Several numerical approaches are available for estimating the effective moisture diffusion coefficient of composites with voids defects by solving a unit cell problem on the RVE model. The existing theoretical models are listed in Table 4. The diffusivity ratio between composites and matrix ($D_{eff}/D_{m}$) is employed to validate the accuracy of the predicting models. Compared with the experimental value (0.47), the results of the FEM model range from 0.52 to 0.57, which is closer to the experimental data. Thus, the FEM model is accurate and effective, which can be selected as the benchmark to evaluate the accuracy of the other theoretical models. As shown in Figure 3, $D_{eff}/D_{m}$ calculated from the Four-phase model and the Self-consistent model have apparent discrepancies with the result of FEM. In contrast, the Porous-matrix model agrees well with FEM when the void content is less than 1%, but the difference increases significantly with the increasing void content.

In this study, a more accurate theoretical model based on the Halpin–Tsai equation is developed to predict the diffusion coefficient of three-component composites containing the carbon fiber, the PA6 matrix and the voids. The classical Halpin–Tsai equation ignores the presence of voids, and it is widely used to predict the transverse water diffusivity of two-component composites [32]. (3)$$\frac{D_{eff}^{'}}{D_{m}}=\frac{1-V_{f}+\psi^{'}\left(1+V_{f}\right)}{1+V_{f}+\psi^{'}\left(1-V_{f}\right)}$$
with (4)$$\psi^{'}=\alpha^{'}\frac{D_{f}}{D_{m}}$$
where *D_f_*, *D_m_* and $D_{eff}^{'}$ are the diffusivity of the fiber, matrix, and composite, respectively, *V_f_* is volume fraction of fibers, $\alpha^{'}$ and $\psi^{'}$ is the moisture concentration and diffusion coefficient gap between the fiber and the matrix, respectively, and $\alpha^{'}$ is defined as follows: $a^{'}=m_{f}r_{f}/m_{m}r_{m}$

To introduce the water absorption parameters of voids to the Halpin–Tsai equation, the three-phase composite model can be divided into two two-phase composite models (submodel-1 and submodel-2). Submodel-1 comprises the matrix and voids, and voids are embedded in the matrix to form the porous matrix phase. In Submodel-2, the porous matrix phase and the fiber consist of the composite. By substituting the parameters of component materials in submodel-1 and submodel-2 into Equation (3), the transverse diffusivity of three-component composites can be expressed as:(5)$$\frac{D_{eff}}{D_{m}}=\frac{1-V_{f}}{1+V_{f}}\cdot \frac{1-V_{v}+\alpha \left(1+V_{v}\right)D_{v}/D_{m}}{1+V_{v}+\alpha \left(1-V_{v}\right)D_{v}/D_{m}}$$
where $D_{v}$ and $V_{v}$ are the diffusivity and volume fraction of voids, respectively, α is diffusion coefficient gap between voids and the matrix.

As shown in Figure 3, $D_{eff}/D_{m}$ determined by the Halpin–Tsai model based on Equation (5) agree well with FEM results. Therefore, the Halpin–Tsai model is accurate and effective, which can rapidly predict the transverse diffusivity of composite with different volume fractions of voids.

As shown in Figure 4, the water absorption curves obey the Fick law, and the moisture absorption ability can be obviously enhanced due to the presence of voids. Water absorption parameters increase linearly as the void content increases, and every 1% increase in the void content results in a 1.76% and 3.1% increase in the water diffusion coefficient and water-absorbed content, respectively. The trend of curve for the water-absorbed content versus void content agrees well with that of carbon/epoxy composites [33].

### 4.2. Residual Stress Evolution

Residual stresses occur in composite structures during the curing process and subsequence service in the hygrothermal environment, which play an important role in composite deformation and mechanical properties. However, the experimental measurement of the residual stresses is often costly and complicated. Alternatively, a numerical simulation method is employed to predict the residual stresses during the process of cooling contraction and hygroscopic expansion, respectively.

As shown in Figure 5a, in the preparation of CF/PA6 laminates by the molding process, the CF/PA6 prepreg is heated to a processing temperature, and subsequently solidified upon cooling from 160 °C to the ambient temperature (about 25 °C). The mismatch in the coefficients of thermal expansion (CET) between the CFs (−0.83 × 10^−6^/°C) and the PA6 matrix (4 × 10^−6^/°C) results in a significant difference in deformation behaviors. Figure 5d describes the changing revolution of residual stress for the matrix and the fiber with the temperature, in the curing process. Through the stress nephogram and the corresponding legend, under the deformation coordination constraint, tensile stress arises in the PA6 matrix, and compressive stress occurs in the CFs. Stress components (S11/S22, S12, Mises stress) increase linearly with the decreasing temperature for the reference points in the CFs and matrix (Figure 5b). In contrast, the average value of stress components for the CFs and matrix is shown in Figure 5c. The average stresses for the CFs and matrix are relatively small after the temperature cooling to 25 °C, with average Mises stress of 1.70 MPa and 1.44 MPa, respectively, which indicates the thermal residual stress is not the main reason for the mechanical damage.

The PA6 matrix is highly sensitive to the moisture under the hygrothermal environment due to the presence of amide groups -CO-NH-, and the absorbed water content can reach 10%. Moreover, the retention rate of mechanical properties for the PA6 matrix decreases exponentially as the water content increases. Thus, water content has deleterious effects on the dimensional stability and mechanical properties in the service environment. The residual stress of hygroscopic expansion in the water absorption process can accelerate mechanical aging.

As shown in Figure 6a, and after CF/PA6 composites’ immersion in the distilled water, the volume of the PA6 matrix expands, while the CFs are not affected by the water. Figure 6d describes the changing revolution of residual stress for the matrix and the fiber with the temperature under the hygrothermal environment. Through the stress nephogram and the corresponding legend, under the deformation constraint, compressive stress arises in the PA6 matrix and tensile stress occurs in the CFs. Figure 6b shows the evolution of stress components for the reference points in the CFs and matrix in the water absorption process. The magnitude of residual stress depends on two factors: water content difference and moisture expansion coefficients. The PA6 matrix exhibits tensile stress before hygrothermal aging, and it is the thermal residual stress in the manufacturing process. In the initial water absorption process, the tensile stress increases from 0.58 MPa to 1.65 MPa. It is explained as follows: the reference point is located in the middle area of the RVE model, and the water diffusion front has not arrived. Tensile stress arises in the middle area due to the water content difference between the internal and external area, resulting in an increase in the stress value. As the water diffuses into the middle area and the water content increases, compressive stress occurs in this area, which can reduce the value of the tensile stress and change the stress from the tension (1.65 MPa) to compression (−1.52 MPa). On the contrary, the tension stress arises in the CFs and the stress value increases from −0.46 MPa to 1.15 MPa, though a small fluctuation occurs in the initial process. The average value of stress components for the CFs and matrix in the water absorption is shown in Figure 6c. The shear stress (S12) is almost unchanged, but the normal stress (S22) of the matrix decreases from 0.43 MPa to −1.19 MPa and that of CFs increases from −0.56 MPa to 1.63 MPa. It is worth noting that the transverse tensile strength of CF/PA6 composites is about 9.5 MPa. Compared with the thermal residual stress, the hygrothermal residual stress is closer to the interfacial strength, which can promote initiation and propagation of cracks in the matrix and interfacial debonding.

To study the effects of voids content on the residual stress in the cooling contraction and hygroscopic expansion process, the same material properties and boundary conditions are applied into the RVE models with the different volume fractions of voids.

Moreover, to avoid duplication and redundancy, only the evolution of Mises stress is selected to quantize the effects of voids content. As shown in Figure 7, the Mises stress decreases significantly from 1.5 MPa to 0.47 MPa in the cooling contraction process, and decreases from 3.77 MPa to 2.88 MPa in the hygroscopic expansion process. It is worth noting that the decrease rate is evident with voids content of 1%, but the reduction trend gradually slows down. This indicates that the stress distribution can be obviously changed, and the presence of voids can greatly weaken the residual stress. The increase in the voids content has slight effects on the decrease of residual stress. However, voids have beneficial effects on the relief or decrease the residual stress to some extent. Before and after hygrothermal aging, the effects of voids on the mechanical properties under the different loading conditions will be discussed in the next section.

### 4.3. Mechanical Performance and Failure Analysis

#### 4.3.1. Voids Effects on Mechanical Performances

Figure 8 plots the strength and elastic modulus as a function of voids content without hygrothermal aging for three loading cases, including transverse tension, compression and shear. The voids content has more deleterious effects on the strength than the modulus under different loading, and the reduction of strength is about two times that of modulus. Quantitatively, voids with a volume fraction of 5% decrease the tensile strength by 32.0% and about 16.1% for tensile modulus. In general, the strength in the transverse direction is dominated by the performance of the matrix and the interface. However, the modulus is depended on the modulus and volume fractions of component materials, and it can be estimated using a rule-of-mixture law. It is important to note that the modulus of CFs (16.54 GPa) is one order of magnitude higher than that of the matrix (2.19 GPa). Voids change the stress distribution and lead to the stress concentration in the matrix, but has a slight influence on CFs. Therefore, the strength is more sensitive than the modulus with different voids contents.

For the models with 5% voids content before hygrothermal aging, the transverse tensile strength ($\sigma_{t}$) suffers the greatest reduction (32.0%), followed by that of the transverse tensile strength ($\sigma_{c}$) with a reduction (29.8%). In contrast, the reduction of transverse shear strength ($\sigma_{s}$) is minimum (14.15%). The reduction of modulus follows the same order: $E_{t}$ (16.1%) > $E_{c}$ (15.5%) > $E_{s}$ (9.24%). Quantitatively, every 1% increase in the void content results in a 6.4%, 5.96% and 2.83% decrease in $\sigma_{t}$, $\sigma_{c}$ and $\sigma_{s}$ respectively. Similarly, every 1% increase in voids content results in a 3.22%, 3.1% and 1.85% decrease in $E_{t}$, $E_{c}$ and $E_{s}$ respectively. This phenomenon can be explained by the angle between crack propagation and loading direction. The angle of crack propagation in the transverse tension, compression and shear is about 90°, 56° and 45°, respectively. This angle is a positive correlation with the reduction rate of mechanical performance.

There is a distinct inflection point in the evolution of strength as shown Figure 9. When voids content is less than 1%, the reduction rate is significantly higher than from 1% to 5%. With the voids content ranging from 1% to 5%, every 1% increase in the void content results in a 3.86%, 2.91% and 2.19% decrease in $\sigma_{t}^{'}$, $\sigma_{c}^{'}$ and $\sigma_{s}^{'}$ respectively. These reductions for the models after hygrothermal aging are lower than those of corresponding unaged models, and the failure mechanism is different. For unaged models, the voids content is the main reason for the mechanical degradation. In contrast, the voids and the water content combined determine the mechanical degradation after the hygrothermal aging. However, the contribution of voids content and water content to the reduction of mechanical performances are different. The decrease in the tensile strength induced by voids content (32.0%) is lower than that of water content (63%), and the negative effects of voids content are diminished. However, the variation trend of modulus is consistent with that of unaged models. After hygrothermal aging, every 1% increase in voids content results in a 3.16%, 2.73% and 2.13% decrease in $E_{t}^{'}$, $E_{c}^{'}$ and $E_{s}^{'}$ respectively.

#### 4.3.2. Progressive Failure Analysis

Figure 10 shows the final damage of RVE models with three different voids content under the transverse tension, compression and shear. It is noted that RVE models have different final damage paths with different voids content. Voids significantly change the stress distribution and cause stress concentration of RVE models. The initial damage is induced around voids and the interface simultaneously, leading to the matrix damage and interfacial debonding, respectively. Then, the cracks propagate to neighbor areas to jointly create the final damage crack.

The final damage patterns of RVE models after hygrothermal aging and mechanical loading are presented in Figure 11. The mechanical properties decrease significantly after hygroscopic saturation, especially for the matrix. The presence of voids can accelerate the matrix damage and interfacial debonding. Therefore, the number and density of cracks increase after hygrothermal aging, compared with unaged RVE models.

## 5. Conclusions

To reveal the effects of voids content on the moisture absorption performances and mechanical properties under the loading of transverse tension, compression and shear, RVE models with different voids contents are established and analyzed. The following conclusions can be drawn:(1)Water diffusion is significantly affected by the presence of voids, which significantly accelerate the water absorption rate and increase the water saturation content. Moreover, the modified Halpin–Tsai equation can effectively predict the transverse diffusivity of composite with different volume fractions of voids.(2)Tensile stress arises for the matrix in the cooling contraction of the molding process, and the stress change from tension to compression after hygrothermal aging. The stress state in CFs is just the opposite of the matrix. Stress distribution can be changed, and the residual stress can be greatly weakened by the presence of voids. Voids have beneficial effects on the relief or decrease the residual stress to some extent.(3)For the unaged composites, voids induce the decrease of mechanical properties due to stress concentration, and every 1% increase in the voids content results in a 6.4% decrease of transverse tensile strength. However, matrix degradation due to the absorbed water content after hygrothermal aging is the dominant factor, and the corresponding rate is 3.86%. This indicates that the negative effects of voids content are diminished, and hygrothermal aging is the dominant reason for the failure mechanism.(4)Compared with the failure images of unaged composites, the number and density of cracks increase after hygrothermal aging, which indicates the presence of voids can accelerate the matrix damage and interfacial debonding.

## Figures and Tables

**Figure 1 polymers-14-00901-f001:**
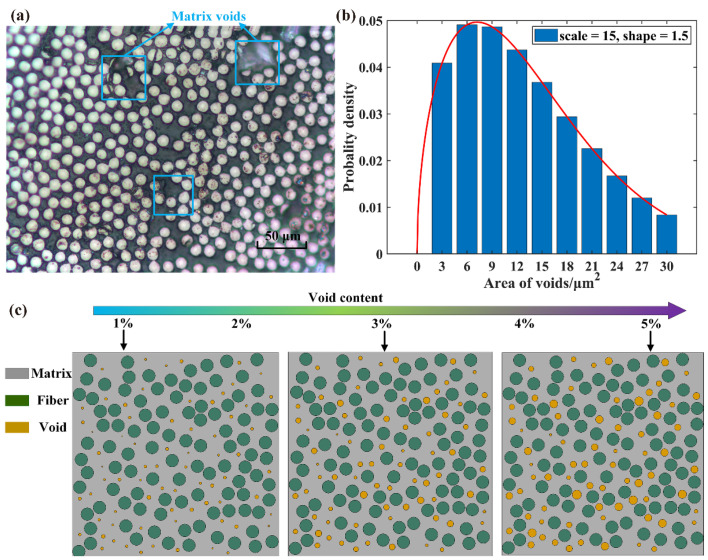
RVE model with different contents of voids. (**a**) metallographic diagram of voids distribution, (**b**) distribution of void size, (**c**) geometric model.

**Figure 2 polymers-14-00901-f002:**
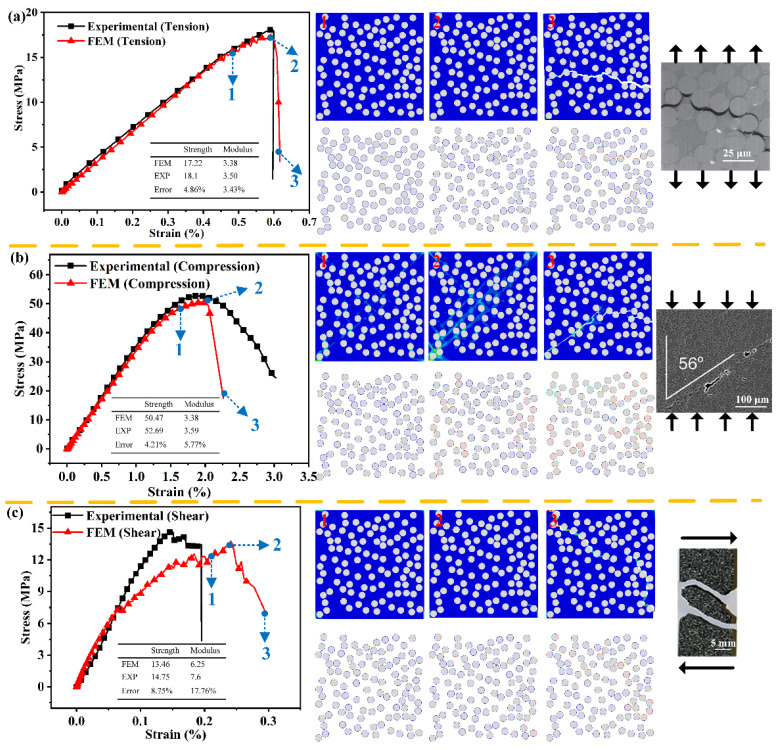
Failure progression in an RVE model under the loading of (**a**) transverse tension, (**b**) transverse compression, (**c**) transverse shearing.

**Figure 3 polymers-14-00901-f003:**
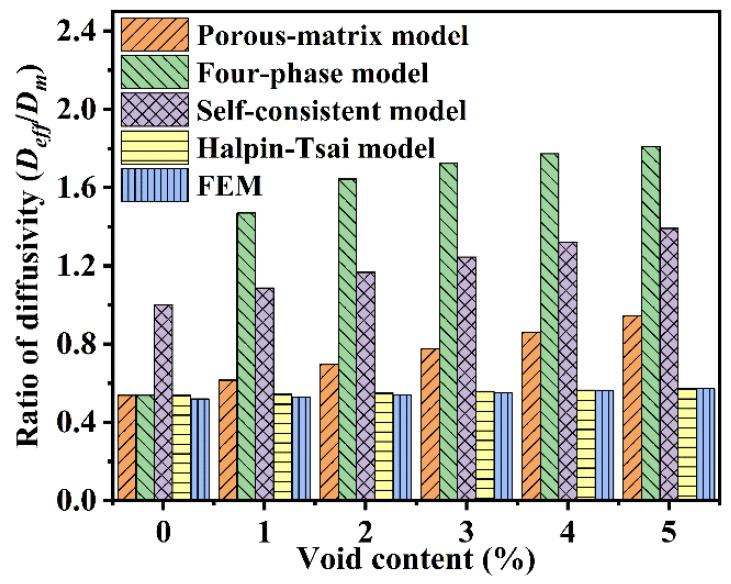
Comparison of prediction models and numerical simulation results.

**Figure 4 polymers-14-00901-f004:**
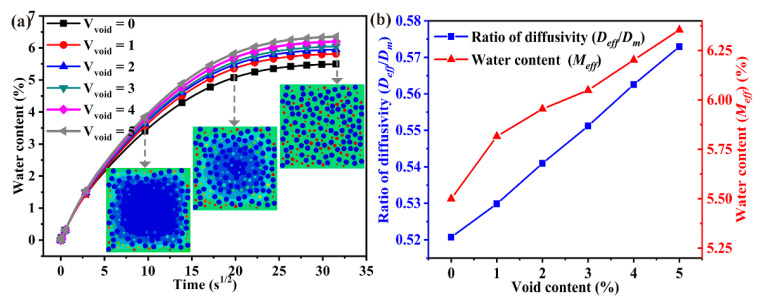
Effect of void content on the water absorption performance, (**a**) water content, (**b**) ratio of diffusivity.

**Figure 5 polymers-14-00901-f005:**
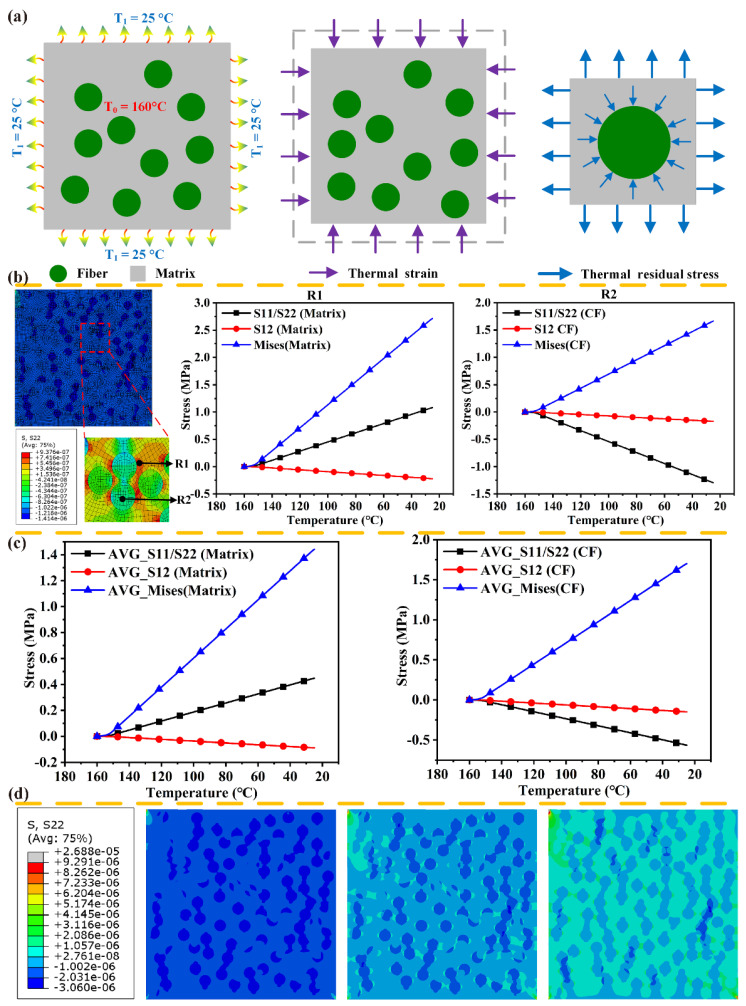
Residual stress of cooling contraction in the molding process, (**a**) schematic diagram, (**b**,**c**) residual stress of component material, (**d**) colored stress patterns.

**Figure 6 polymers-14-00901-f006:**
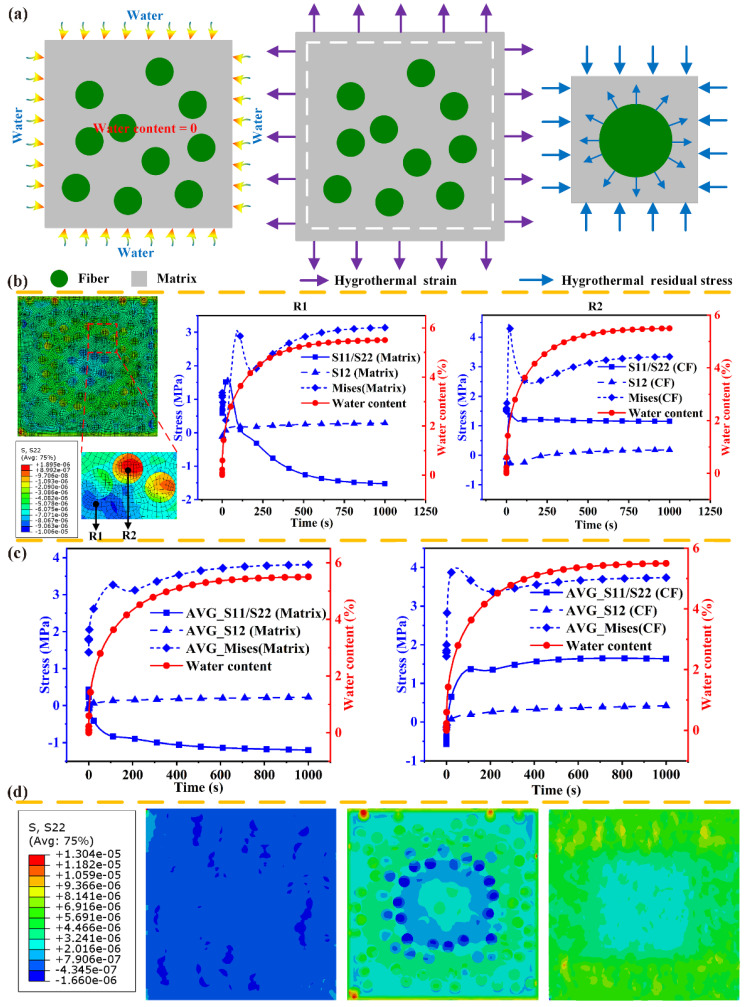
Residual stress of hygroscopic expansion in water absorption process, (**a**) schematic digram, (**b**,**c**) residual stress of component material, (**d**) colored stress patterns.

**Figure 7 polymers-14-00901-f007:**
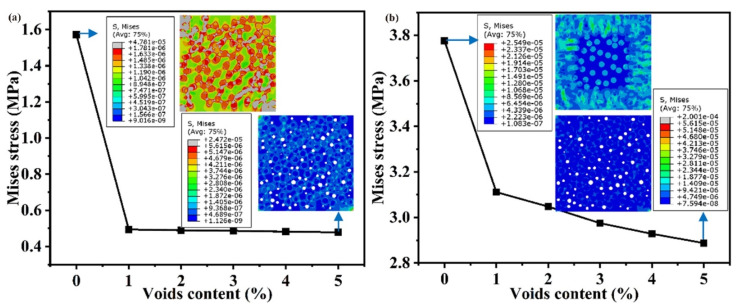
Effect of voids content on the residual stress, (**a**) Mises stress of cooling contraction, (**b**) Mises stress of hygroscopic expansion.

**Figure 8 polymers-14-00901-f008:**
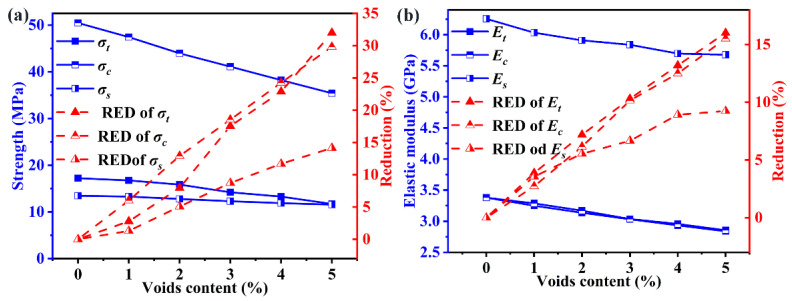
Effect of voids content on the mechanical performance for unaged samples, (**a**) strength, (**b**) elastic modulus.

**Figure 9 polymers-14-00901-f009:**
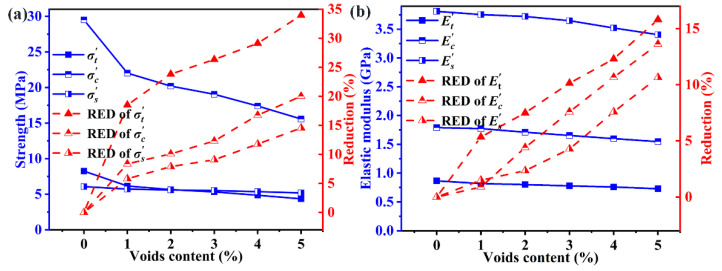
Effect of voids content on the mechanical performance for hygroscopic saturated samples, (**a**) strength, (**b**) elastic modulus.

**Figure 10 polymers-14-00901-f010:**
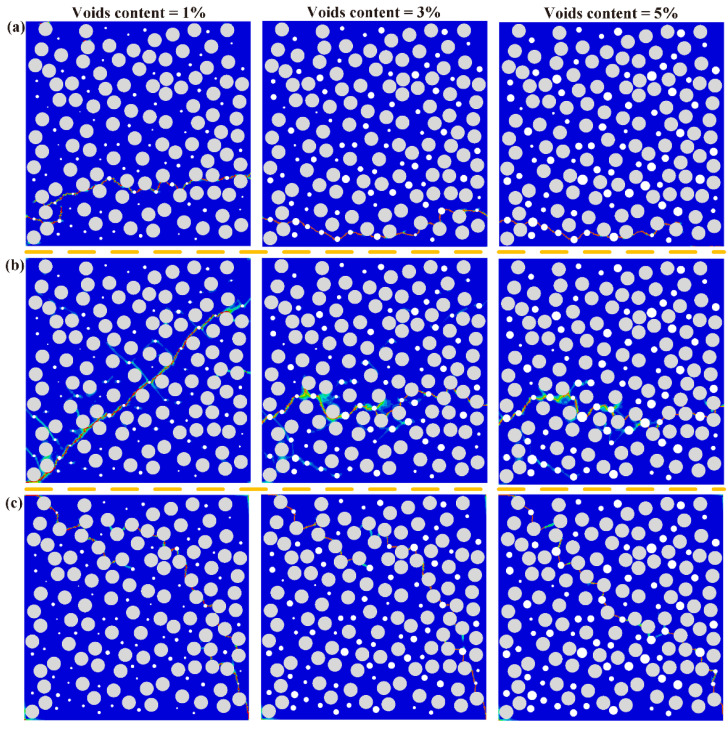
Colored failure patterns of unaged samples in the (**a**) transverse tension, (**b**) transverse compression, (**c**) transverse shear simulation.

**Figure 11 polymers-14-00901-f011:**
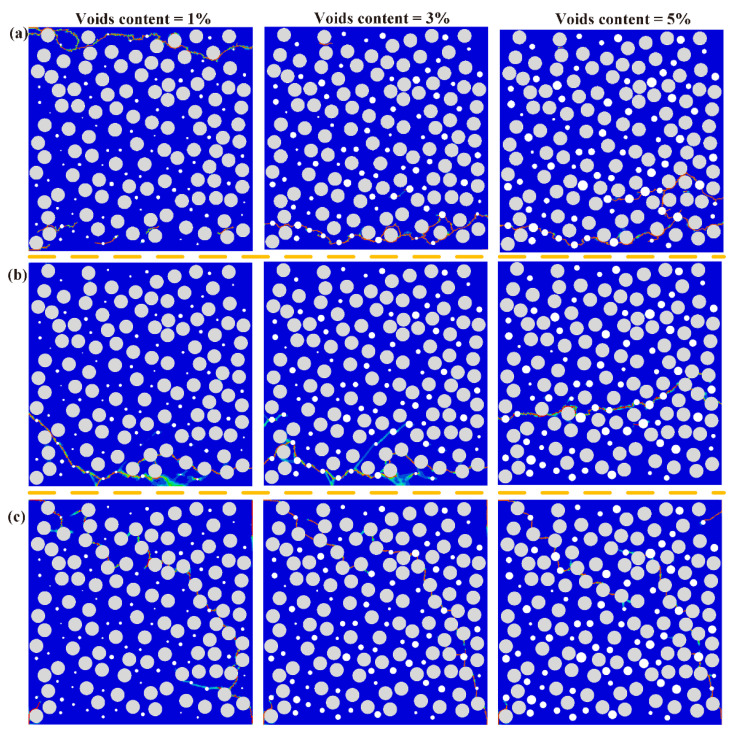
Colored failure patterns of hygroscopic saturated samples in the (**a**) transverse tension, (**b**) transverse compression, (**c**) transverse shear simulation.

**Table 1 polymers-14-00901-t001:** Water absorption parameters for component materials in the CF/PA6 composites immersed at 50 °C water bath.

	Matrix	Carbon Fiber	Voids	Composite
$\mathrm{Diffusivity}\;(10^{-6}{\mathrm{mm}}^{2}/s$)	$D_{m}$	$D_{f}$	*D_v_*	$D_{eff}$
	4.64	0	46.4	2.16
Water content (%)	$M_{m}$	$M_{f}$	$M_{v}$	$M_{eff}$
	9.4	0	23.5	5.26

**Table 2 polymers-14-00901-t002:** Mechanical properties for CF/PA6.

Fiber	$E_{2}^{f}(\mathrm{GPa})$	$\mu_{23}^{f}$	$\alpha^{f}(10^{-6}/{}^{\circ}C)$	$\beta^{f}$	$\rho^{f}(\mathrm{kg}/m^{3})$
	16.54	0.25	−0.83	0	1810
Matrix	$E^{m}(\mathrm{GPa})$	$\mu^{m}$	$\sigma_{yt}(\mathrm{MPa})$	$\sigma_{yc}(\mathrm{MPa})$	$\alpha^{m}(10^{-6}/{}^{\circ}C)$
	2.19	0.34	25	50	4
	$\beta^{m}$	$\rho^{m}(\mathrm{kg}/m^{3})$			
	0.1	1080			
Interface	$K_{n}^{0}(N/{\mathrm{mm}}^{3})$	$K_{s}^{0}(N/{\mathrm{mm}}^{3})$	$K_{t}^{0}(N/{\mathrm{mm}}^{3})$	$t_{n}^{0}(\mathrm{MPa})$	$t_{s}^{0}(\mathrm{MPa})$
	$3.13\times 10^{4}$	$5.0\times 10^{4}$	$5.0\times 10^{4}$	17.11	40.67
	$t_{t}^{0}(\mathrm{MPa})$	$G_{n}^{c}(N/\mathrm{mm})$	$G_{s}^{c}(N/\mathrm{mm})$	$G_{t}^{c}(N/\mathrm{mm})$	
	40.67	0.22	0.23	0.23	

**Table 3 polymers-14-00901-t003:** Mechanical properties of CF/PA6 composites in the transvers tensile, compression and shear test.

	Transverse Tension	Transverse Compression	Transverse Shear
Strength (MPa)	18.10 ± 1.87	52.69 ± 3.25	14.75 ± 1.28
Modulus (GPa)	3.50 ± 0.38	3.59 ± 0.16	7.60 ± 0.23

**Table 4 polymers-14-00901-t004:** Several theoretical models for calculating the effective diffusion coefficient of composites containing voids [20].

Model	Expression
Porous-matrix model	$\frac{D_{eff}}{D_{m}}=\left[1+2\alpha_{mv}V_{v}\frac{\left(D_{v}/D_{m}\right)-1}{\left(D_{v}/D_{m}\right)+1}\right]\frac{1-V_{f}}{1+V_{f}}$ with $\alpha_{mv}=\frac{\rho_{water}}{M_{m}\rho_{m}(1-v_{void})}$
Four-phase model	$\frac{D_{eff}}{D_{m}}=\frac{\left(1-kV_{f})(1+k)+\Phi (1+kV_{f})(k-1\right)}{\left(1+kV_{f})(1+k)+\Phi (1-kV_{f})(k-1\right)}$ with $k=\left(V_{v}+V_{f}\right)/V_{f}$$,\;\Phi =\alpha_{mv}(D_{v}/D_{m})$
Self-consistent model	$\frac{D_{eff}}{D_{m}}=\frac{1}{2(1+k)}\left(\lambda +\sqrt{4\frac{D_{v}}{D_{m}}(k^{2}-1)+\lambda^{2}}\right)$ with $\lambda =\frac{D_{v}}{D_{m}}\left(1+k\left(2V_{v}-1\right)\right)+\left(1+k\left(1-2V_{v}\right)\right)$

## Data Availability

The data presented in this study are available on request from the corresponding author.

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
