# Peer review of "Effect Mechanism and Simulation of Voids on Hygrothermal Performances of Composites"

_polymers, 2022, doi:10.3390/polym14050901_

Round 1

Reviewer 1 Report

The manuscript entitled as ‘’ Effect mechanism and simulation of voids on hygrothermal performances of composites’’  deals with the use of the computational modellling considering hygrothermal aging to study voids by representative volume element methodology. Overall, this manuscript lacks the novelty and provides little new insight about the RVE for study voids.. Further, the data analysis presented in the current manuscript should be drastically improved. I therefore recommend revision before publication in Polymers.

Although the introduction is well written, I can not see any convincing arguments to justify the representative volume element with order voids investigation in present study. Also, literature reports vastly discuss the use  RVE model with a single void and a stress concentration factor was introduced to quantify the effect of void volume fraction, void orientation, shape density and associated void defects to illustrate mechanical properties of the composites. This should be imperatively included in the paper.

There is no argument to justify the increased water saturation content and absorption rate  results.

It would be better to rewritten the conclusions, since they are not supported by the current data. Also, avoid using number to describe the conclusion.

 Why the matrix mechanical properties greatly decrease  after hygroscopic saturation? And how the authors could associate the presence of voids accelerating the matrix damage and interfacial debonding compared with unaged RVE models?

The English language of the results and discussion section needs to be improved.

Reviewer 2 Report

I found the manuscript "Effect mechanism and simulation of voids on hygrothermal performances of composites" by Zhu Liu et al. interesting and suitable for MDPI Polymers scope. However, I have the following questions for manuscript authors.

  1. What is the unit of Area of voids in Fig. 1b? Also, please make the scalebar in Fig. 1a more visible as it is barely noticeable.
  2. Was there only one sample for each type of mechanical testing: tension, compression, and shear (Fig. 2)? Can you provide these data for several replicates and, therefore, provide mean values with standard deviation or error on the plots?
  3. I assume that the 2x3 array of images in the center of each panel of Fig. 2 results from the RVE model, but please explain "blue" and "white" images in rows, stress value for each column.
  4. Please provide scale bars for each microscopy image (I assume they are microscopy images) of experimental samples in Fig. 2 with the particular microscopy method used.
  5. First paragraph in 4.1. section (lines 174-177) should be removed as part of the manuscript template.
  6. What is the difference between stress maps in Fig. 5d? Is it temperature, or do they have different color schemes? I could not find it in the text (as there is no reference to Fig. 5d) or in Fig. 5 caption. If it is a temperature or any other parameter, please provide it in the left corner or right above the image itself.
  7. I also have recurring questions for each stress pattern you provide. What are units of these color bars? What does "Avg: 75%" mean?
  8. There is a typo in lines 349, 350, change "REV model" to "RVE model".

    Overall, I think that this manuscript should be considered for publication after major revision.
